# Protein-Based Mechanism of Wheat Growth Under Salt Stress in Seeds Irradiated with Millimeter Waves

**DOI:** 10.3390/ijms26010253

**Published:** 2024-12-30

**Authors:** Setsuko Komatsu, Rachel Koh, Hisateru Yamaguchi, Keisuke Hitachi, Kunihiro Tsuchida

**Affiliations:** 1Faculty of Environment and Information Sciences, Fukui University of Technology, Fukui 910-8505, Japan; 2Department of Medical Technology, Yokkaichi Nursing and Medical Care University, Yokkaichi 512-8045, Japan; h-yamaguchi@y-nm.ac.jp; 3Center for Medical Science, Fujita Health University, Toyoake 470-1192, Japan; hkeisuke@fujita-hu.ac.jp (K.H.); tsuchida@fujita-hu.ac.jp (K.T.)

**Keywords:** proteomics, wheat, salt stress, millimeter-wave irradiation, reactive oxygen species-scavenging system

## Abstract

Wheat is one of the most extensively grown crops in the world; however, its productivity is reduced due to salinity. This study focused on millimeter wave (MMW) irradiation to clarify the salt-stress tolerance mechanism in wheat. In the present study, wheat-root growth, which was suppressed to 77.6% of the control level under salt stress, was recovered to the control level by MMW irradiation. To reveal the salt-stress tolerance mechanism of MMW irradiation on wheat, a proteomic analysis was conducted. Proteins related to cell cycle, proliferation, and transport in biological processes, as well as proteins related to the nucleus, cytoskeleton, and cytoplasm within cellular components, were inversely correlated with the number of proteins. The results of the proteomic analysis were verified by immunoblot and other analyses. Among the proteins related to the scavenging reactive-oxygen species, superoxide dismutase and glutathione reductase accumulated under salt stress and further increased in the MMW-irradiated wheat. Among pathogen-related proteins, pathogenesis-related protein 1 and the Bowman–Birk proteinase inhibitor decreased under salt stress and recovered to the control level in the MMW-irradiated wheat. The present results indicate that MMW irradiation of wheat seeds improves plant-growth recovery from salt stress through regulating the reactive oxygen species-scavenging system and the pathogen-related proteins. These genes may contribute to the development of salt-stress-tolerant wheat through marker-assisted breeding and genome editing.

## 1. Introduction

Wheat is one of the most extensively grown crops in the world [1]. However, climate change poses major challenges to wheat agricultural production both globally and regionally [2]. For these reasons, the demand for wheat production inevitably increases around the world. To improve wheat production in salt-damaged arid and semi-arid regions, it is necessary to clarify the adaptation mechanisms of wheat to salt damage [3]. Increased salinity has significant adverse effects on the biochemical, physiological, and morphological characteristics of plants, causing a decrease in the germination rate, photosynthesis, transpiration, disturbing the expected metabolic processes of plants, and then negatively affecting crop productivity [4]. Clarifying these mechanisms by which wheat copes with unexpected salinity, is crucial for developing new salt-tolerant wheat cultivars.

Several molecular, biochemical, and physiological changes are involved in the response of crop plants to salt stress [5], which are classified into three main categories [6]: osmotic tolerance [7], ion exclusion [3], and tissue tolerance to high salt concentrations present in leaves. Osmo-protective mechanisms include glycine-betaine accumulation, reactive-oxygen species (ROS) scavenging, and hormonal modulation in wheat [8]. The main genes involved in salt exclusion are salt overlay sensitive (SOS) homologs, such as *TaSOS1* and *TdSOS1* in wheat [9]. The NHX (Na^+^/H^+^ antiporter) group is responsible for the vacuolar sequestration of Na^+^, and four proteins encoding NHX genes, including *TaNHX1*, TaNHX2, TaNHX3, and *TaNHX4*-*B*, were characterized in wheat vacuole [10]. Fifty-six HAK (high-affinity potassium) transporters induce salt tolerance in wheat [11]. In addition to ionic mechanisms, salt tolerance mechanisms mediated by phytohormones, such as jasmonic acid, salicylic acid, and abscisic acid, are well documented in wheat [12]. The mechanism of salt tolerance in wheat needs to be elucidated in more detail.

To clarify the mechanism of salt-stress tolerance, we focused on electromagnetism. In electromagnetism, millimeter waves (MMW) are located at the overlap between the infrared and microwave regions. Their radio frequencies are between 30 and 300 GHz, and the wavelengths are between 10 and 1 mm [13]. With long wavelengths, which are not permeable to water and pose minimal risks to human health, MMW irradiation is a suitable technology for the environment, which has dynamic effects on organisms [14]. Electromagnetism is linked to biochemical processes and functions within organisms. Within the electromagnetic spectrum, low-intensity microwaves had no effect on plant growth, but high-intensity irradiation increased the seed-germination rate of napa cabbage [15]. In brown rice, MMW irradiation enhanced seed germination and polyphenol content, while it decreased gamma-aminobutyric acid [16]. Wheat seeds irradiated with MMW developed into plants with better yields compared to untreated ones [17]. MMW irradiation of wheat seeds at an early stage improved not only seed germination rate [18], but also plant growth and grain yield [19]. These findings indicate that biochemical processes are necessary for electromagnetically induced functional expression in wheat.

Furthermore, MMW radiation improved plant growth and stress tolerance, including tolerance to flooding in soybean [20], chickpea [21], and wheat [22]. These results indicated that MMW irradiation could be an effective method to promote plant growth and stress tolerance in crops. To clarify the salt-stress tolerance mechanism of MMW irradiation on wheat, morphological analysis was carried out using irradiated seeds. Because MMW irradiation induced structural changes in proteins, proteomic techniques were performed using roots, whose growth in irradiated wheat was improved under salt stress. This was performed using nano-liquid chromatography (LC), combined with mass spectrometry (MS), to explore the responsible mechanisms for the positive effects of MMW on wheat growth under salt stress. The proteomic results were then confirmed by protein accumulation and other analyses using immunoblot and polymerase-chain reaction (PCR) techniques.

## 2. Results

### 2.1. Root Growth of Wheat Under Salt Stress Was Improved by MMW Irradiation

To examine the positive effect of MMW irradiation on wheat under salt stress, the morphological changes in the irradiated wheat were analyzed. Both wheat seeds irradiated with 20 mW of MMW for 20 min and unirradiated wheat were used. Three days after sowing, the wheat was treated with 100 mM NaCl for 2 days. Nontreated plants were used as the control (Appendix A). In the absence of irradiation, salt stress decreased the fresh weight and length of wheat roots to 77.6% of the control (Appendix A). This suppression was restored to the control level in irradiated wheat under stress (Figure 1). However, based on statistical analysis using a *t*-test, no significant changes were observed in fresh weight and length of leaf under salt stress in both unirradiated and irradiated wheat (Figure 1). Based on the significant morphological changes, wheat roots were collected and subjected to proteomic analysis.

### 2.2. Salt Stress-Affected Wheat Proteins, Which Were Restored at the Protein Level by MMW Irradiation

To investigate the protein changes induced by MMW irradiation in wheat growth under salt stress, MS-based proteomics was carried out using roots. Four types of treatments, which were irradiated/unirradiated and salt stress/nontreated, were conducted (Appendix A). The proteins extracted from the roots were enriched, reduced, alkylated, digested, and analyzed using nanoLC-MS/MS. The relative protein abundance of unirradiated or irradiated wheat was compared between salt stress treated and nontreated groups. A total of 8948 proteins were detected by MS-based proteomics (Appendix A). The proteomic results of all 12 samples from the four different groups were compared using principal-component analysis, which showed unique accumulation patterns of proteins across all groups (Figure 2). The principal-component analysis showed that salt stress largely affected the wheat proteins, but this change was mitigated at the protein level by MMW irradiation, even if it was under stress (Figure 2).

In unirradiated wheat roots, the abundance of 477 proteins was differentially altered under salt stress, with fold changes of ≥1.5 and ≤0.67 compared with nontreated wheat. Among the 477 proteins, 329 increased and 148 decreased under salt stress compared to nontreated wheat (Figure 3A and Appendix A). Additionally, the abundance of 139 proteins was differentially altered with fold change ≥1.5 and ≤0.67 in the roots of MMW irradiated wheat under salt stress compared with nontreated ones. Among the 139 proteins, 86 increased and 53 decreased with salt stress compared to nontreated wheat (Figure 3B and Appendix A). Functional categories were obtained using gene-ontology analysis. In the biological processes, the proteins related to the cell cycle, proliferation, and transport were reversely changed with the number of proteins. In the cellular component category, proteins related to the nucleus, cytoskeleton, and cytoplasm were inversely correlated with the number of proteins (Figure 3). Meanwhile, in the molecular processes, there was no significant difference between the unirradiated and irradiated wheat under salt stress compared to control (Appendix A). Based on the proteomic results, these identified proteins were further characterized by immunoblot and PCR analyses.

### 2.3. Protein Accumulation Was Altered Under Salt Stress in Wheat Irradiated with MMW, as Determined Using Immunoblot Analysis

Because proteomic analysis revealed changes in the amounts of ROS-scavenging and pathogenesis-related proteins, the changes in protein amounts were confirmed by immunoblot analysis (Figure 4 and Figure 5). Proteins were extracted from the root and leaf samples of wheat, whose seeds were irradiated with or without MMW, under nontreated or salt-stress. Proteins were separated on SDS-polyacrylamide gel by electrophoresis. Coomassie-brilliant blue staining pattern was used as a loading control (Appendix A).

To confirm the changes in ROS-scavenging-related proteins, the accumulation of superoxide dismutase (SOD) 1 (Cu/Zn SOD), glutathione reductase (GR), and ascorbate peroxidase (APX) was analyzed by immunoblot analysis (Figure 4 and Appendix A). SOD 1 and GR accumulated in the root and leaf under salt stress, while they further accumulated in the root of wheat irradiated with MMW (Figure 4A,B). The abundance of cytosolic APX decreased with salt stress; however, there was no change with irradiation. The abundance of mitochondrial APX was not altered by any treatment (Figure 4C,D).

To confirm the changes in pathogen-related proteins, the accumulation of pathogenesis-related protein 1, chitinase, thaumatin, and the Bowman–Birk proteinase inhibitor was analyzed using immunoblot analysis (Figure 5 and Appendix A). The abundances of pathogenesis-related protein 1, chitinase, and Bowman–Birk proteinase inhibitor decreased, while thaumatin increased in the root under salt stress (Figure 5). The suppressed abundance of pathogenesis-related protein 1 and Bowman–Birk proteinase inhibitor was restored in the wheat irradiated with MMW (Figure 5A,D).

### 2.4. Gene Expression Was Analyzed Under Salt Stress in Wheat Irradiated with MMW Using PCR Technique

To better uncover the alterations in nuclear-associated proteins caused by MMW irradiation under salt stress, the expression levels of gene-encoding nucreoporin (accession number A0A3B6HV94 in Appendix A) was analyzed in both samples, with and without MMW irradiation, under salt stress, using PCR technique. Total RNAs extracted from the root and the leaf of wheat were used to synthesize cDNA. PCR analysis was performed for the gene expression analysis of *nucleoporin* and *18S rRNA* (Figure 6A,B). The gene-expression levels of *18S rRNA* as an internal control and *nucleoporin* did not change with or without MMW irradiation, regardless of the presence or absence of salt (Figure 6).

### 2.5. Starch Content Decreased Under Salt Stress in Wheat Irradiated with MMW

To confirm the alteration of starch synthetase, whose abundance significantly increased under salt stress compared to the nontreated plants (Appendix A), the starch content in the roots and leaves was measured (Figure 7). Three-day-old wheat was treated with or without salt for 2 days, after which the starch content in the roots and leaves was measured (Figure 7). The starch content significantly decreased under salt treatment compared to the control; however, there was no change between the MMW-irradiated and unirradiated wheat (Figure 7).

## 3. Discussion

### 3.1. MMW Irradiation Positively Affects Wheat Growth Under Salt Stress

The application of microwave technology has attracted attention mainly in the agri-food sector [23]. However, in soybean, MMW irradiation improved the recovery of seedlings from flooding stress and positively regulated glycolysis and ROS scavenging pathways [20]. In chickpea, MMW irradiation regulated leaf photosynthesis and root fermentation, improving plant growth recovery from flooding [21]. In wheat, MMW irradiation helped plant-growth recovery from flooding through the regulation of cell organization, glycolysis, ROS scavenging, and auxin contents [22]. Additionally, MMW irradiation significantly enhanced wheat-root growth under flooding through the alterations of proteins in mitochondria, endoplasmic reticulum, vacuole, and plasma membrane [24], as well as the ubiquitin-proteasome system and mRNA-expression [25]. These results suggest that MMW irradiation had a positive effect on crops against flooding stress. In this study, the irradiation time and the oscillation power were fixed at 20 mW for 20 min, respectively, as these conditions were more effective for wheat [22]. As a result, without irradiation, wheat-root growth was suppressed to 77.6% of the control under salt stress. However, this suppression was restored to the control level in irradiated wheat (Figure 1). In this experiment, no nutrients were added, so even though the roots grew better, the nutrients were not absorbed, so there may have been no effect on the above-ground parts. The result of this study, as well as previous findings, suggests that MMW irradiation is a useful tool for improving the stress tolerance of crops not only against flooding stress but also against salt stress.

### 3.2. MMW Irradiation Regulates the ROS Scavenging Process in Wheat Under Salt Stress

Plants have integrated ROS-detoxification mechanisms, which involve antioxidant enzymes such as SOD, GR, APX, catalase, dehydroascorbate, and monodehydroascorbate reductase, as well as nonenzymatic antioxidants such as reduced glutathione, ascorbic acid, carotenoids, phenols, flavonoids, and α-tocopherol [26]. The application of plant-essential oils indicated a potential alternative strategy to enhance salt tolerance and improve the growth quality of durum wheat seedlings, thereby activating antioxidant defense and increasing ROS scavenging [27]. In wheat seedlings under salt stress, priming with chitosan nanoparticles reduced ion toxicity by upregulating antioxidant mechanisms such as catalase, peroxidase, APX, and SOD, along with the contents of photosynthetic pigments, tannins, flavonoids, and protein [28]. In this study, proteins and proteoforms related to antioxidant enzymes non-enzymatic antioxidants were identified using proteomic analysis; they increased with salt stress and further increased with MMW irradiation under stress (Appendix A). These results align with previous studies, suggesting that MMW irradiation may confer salt stress tolerance to wheat by enhancing the ROS-scavenging system and accelerating the detoxification process.

In wheat, two well-known salt tolerance pathways are sodium efflux via the high-affinity potassium transporter (HKT) and ROS homeostasis regulated by Similar to RCD-One (SRO) [29]. *TaSRO1*, which is encoding a poly (ADP ribose) polymerase domain protein, regulated salt tolerance in wheat through maintaining ROS homeostasis [30]. The ectopic expression of *TaDWF4 (brassinolide synthesis gene)* or *TaBAK1 (brassinolide signaling gene)* balanced ROS levels in roots and enhanced salt tolerance in wheat by removing ROS [31]. *TaBZR1 (brassinazole-resistant 1)* improved salt-stress tolerance through activating some genes involved in ROS scavenging and abscisic-acid biosynthesis in wheat [32]. In this study, among the proteins and proteoforms related to ROS scavenging, the SOD 1 and GR levels increased under salt stress and further enhanced in the MMW-irradiated wheat (Figure 4). Because *TaSOD2* overexpression in wheat increased the SOD activity and improved tolerance to salt/oxidative stress [33], SOD might be one of the important candidates to improve salt tolerance in wheat.

### 3.3. MMW Irradiation Regulates Pathogen-Related Proteins in Wheat Under Salt Stress

A large number of functional factors, which play a role in abiotic stress response, were isolated, including components of the SOS pathway, kinases and phosphatases, transcription factors, ion transporters, and the abscisic acid pathway [34,35,36]. Pathogenesis-related proteins are involved in salicylic acid-mediated disease resistance responses [37,38]. Pathogen-related proteins, including glucanase, thaumatin, and chitinase, inhibit microbial growth through their enzymatic activities [39,40]. They also function under abiotic stress condition: for example, AtPR protein played a role in seed germination under salt stress [41]. Similarly, AtPR1, AtPR3 (chitinase), and AtPR5 (thaumatin) functioned in response to drought stress [42]. Chitinase was involved in various abiotic stress responses in plants, including wounding, osmotic, heavy metal, cold, and salt stresses [43]. PR-10 in spinach and peanut plays a role in stress-signaling pathways [44,45]. Recombinant expression of *TaTLP2-B* in *Saccharomyces cerevisiae* provided significant tolerance to heat, cold, salt, and osmotic stresses [46]. These proteins and proteoforms were also identified by proteomic analysis in wheat irradiated with MMW under salt stress (Appendix A). These findings indicate that pathogenesis-associated proteins not only function to tolerate biotic stresses but also to tolerate abiotic stresses, such as salinity stress.

In this study, the abundance of PR-1 and the Bowman–Birk proteinase inhibitor decreased in the roots of wheat under salt stress; however, it significantly increased with MMW-irradiated wheat (Figure 5). The OsPR-1a protein/sperm coating protein improved the tolerance against abiotic stress [47,48]. The expression of *TaPR-1-1* was induced by salinity, freezing, and osmotic stresses [49]. The Bowman–Birk protease inhibitors in wheat, such as *wali3*, *wali5*, and *wali6*, were induced by wounding or toxic metal stress [50]. The Bowman–Birk proteinase inhibitors were involved in salt-stress tolerance in wheat [51]. This study, taken together with previous findings, suggests that MMW irradiation improves wheat growth under salt stress via accumulation of PR-1 and Bowman–Birk protease inhibitors.

## 4. Materials and Methods

### 4.1. Plant Materials and Salt Treatment

The seeds of wheat (*Triticum aestivum* L. cultivar Nourin 61) were irradiated with MMW at 20 mW for 20 min. The workflow of MMW irradiation was described in previous studies [23,24] (Appendix A). Briefly, a Gunn oscillator was used as an MMW source, with a frequency range of 79 to 115 GHz and an output power of 7 to 80 mW. The antenna pattern of the horn antenna had an aperture angle of 17 degrees on each side. By placing a 5 cm diameter petri dish containing wheat seeds at 15 cm from the horn antenna, the MMW radiation area completely covered the dish. The irradiation time and the oscillation power were fixed at 20 mW for 20 min, respectively, as these conditions were more effective for wheat [22].

After MMW irradiation, a total of 20 seeds were sown evenly in 400 mL of silica sand in each seedling case. Afterward, they were cultivated for 3 days at 25 °C under white-fluorescent light (160 µmol m^−2^ s^−1^, 12 h light period/day). Furthermore, based on previous studies, a growth-inhibiting concentration of NaCl for wheat (100 mM) was used [52]. In this study, to induce salt stress, 100 mM NaCl were supplied for 2 days. As a control, plants were given only water. For morphological analysis, roots and leaves were collected. For proteomic analysis, roots were collected. For other biological and molecular analyses, roots and leaves were collected. Three independent experiments were conducted as biological replications for every experiment, meaning that the seeds were sown on different days. Samples from four groups were collected: wheat with or without salt stress and with or without MMW irradiation (Appendix A).

### 4.2. Protein Extraction and Concentration Measurement

A portion (500 mg) of samples was ground in 500 µL of lysis buffer containing 50 mM Tris-HCl, 150 mM NaCl, 0.1% SDS, 1% Nonidet-P40, and protease inhibitor (Nakalai Tesque, Kyoto, Japan) using a mortar and pestle. The suspension was centrifuged at 16,000× *g* for 10 min at 4 °C. Detergent was removed from the supernatant using a Pierce Detergent Removal Spin Column (Pierce Biotechnology, Rockford, IL, USA). Proteins were incubated with a Protein Assay Kit (Bio-Rad, Hercules, CA, USA) [53] for 5 min, and absorbance was measured at 595 nm. Protein concentrations were calculated using a standard carve of bovine-serum albumin. This sample was used for proteomics analysis and immunoblot analysis.

### 4.3. Protein Enrichment, Reduction, Alkylation, and Digestion

The quantified protein (100 µg) was adjusted to a final volume of 100 µL. The proteins were concentrated, reduced, alkylated, and digested using previous methods [54]. In brief, 400 µL of methanol, 100 µL of chloroform, and 300 µL of water were mixed in the sample. The suspension was centrifuged at 16,000× *g* for 10 min. The upper phase was discarded, and 300 µL of methanol was added to the lower phase, followed by centrifugation at 16,000× *g* for 10 min. The pellet was collected as the soluble fraction and resuspended in 50 mM ammonium bicarbonate. The proteins were reduced with 50 mM dithiothreitol for 30 min at 56 °C and alkylated with 50 mM iodoacetamide for 30 min at 37 °C, in the dark. Then, the proteins were digested with trypsin (Fujifilm Wako Pure Chemical, Osaka, Japan) at an enzyme/protein ratio of 1:100 for 18 h at 37 °C. Peptides were desalted on a MonoSpin C18 Column (GL Sciences, Tokyo, Japan) and acidified with 1% trifluoroacetic acid.

### 4.4. Protein Identification Using LC-MS/MS

The conditions of LC (EASY-nLC 1000; Thermo Fisher Scientific, San Jose, CA, USA) and MS (Orbitrap Fusion ETD MS; Thermo Fisher Scientific) were described in a previous study [55] (Appendix A).

### 4.5. Analysis of MS Data

The MS/MS searches were carried out using the MASCOT (version 2.6.2; Matrix Science, London, UK) and SEQUEST HT search algorithms against the *Arabidopsis Thaliana* (UniProtKB TaxID = 3702) (version 2021-02) and *Triticum aestivum* (SwissProt TaxID = 4565) (version 2021-02) using Proteome Discoverer (version 2.4; Thermo Fisher Scientific). The workflow was described in a previous study [20] (Appendix A).

### 4.6. Differential Analysis of Proteins Using MS Data

Label-free quantification using precursor-ion quantifier nodes and principal-component analysis were performed with Proteome Discoverer. For the differential analysis of the relative abundance of peptides and proteins between samples, the free software PERSEUS (version 1.6.15.0) [56] was used. The workflow was described in a previous study [20] (Appendix A).

### 4.7. Immunoblot Analysis

Quantified protein (10 µg) was added to SDS-sample buffer containing 60 mM Tris-HCl (pH 6.8), 2% SDS, 5% dithiothreitol, 10% glycerol, and bromophenol blue [57] (Bio-Rad). Proteins were separated on a 10% SDS-polyacrylamide gel by electrophoresis. Coomassie-brilliant blue staining was used as a loading control. Proteins were transferred to a polyvinylidene difluoride membrane using a semi-dry transfer blotter. The blotted membrane was blocked with Bullet Blocking One regent (Nacalai Tesque) for 5 min and cross-reacted with the primary antibodies at an antibody/buffer ratio of 1:1000 for 30 min. As the primary antibodies, the following were used: anti-ascorbate peroxidase (APX) [58], glutathione reductase (GR) (Agrisera, Vännäs, Sweden), superoxide dismutase (SOD) (Proteintech, Rosemont, IL, USA), pathogenesis-related protein 1 [59], chitinase [60], thaumatin [61], and Bowman–Birk proteinase inhibitor [62] antibodies. After reaction, the membrane was cross-reacted with anti-rabbit IgG conjugated with horseradish peroxidase (Bio-Rad) as the secondary antibody for 30 min. After the reaction, signals were detected using a TMB Membrane Peroxidase Substrate Kit (Seracare, Milford, MA, USA). The integrated density of the bands was calculated using Image J software (version 1.53e with Java 1.8.0_172; National Institutes of Health, Bethesda, MD, USA).

### 4.8. RNA Extraction, cDNA Synthesis, and PCR Analysis

Total RNA was isolated with the RNeasy Plant Mini Kit (Qiagen, Venlo, The Netherlands) according to the protocol of the manufacturer. A portion (500 mg) of samples was flash-frozen in liquid nitrogen and ground to a powder using a mortar and pestle. First-strand cDNA was synthesized from total RNA (1 μg) with the iSuperscript Reverse Transcription Supermix (BioRad). Gene-specific primers were constructed with Primer3Plus software [63] (https://www.bioinformatics.nl/cgi-bin/primer3plus/primer3plus.cgi/ (accessed on 1 December 2024)) and used to amplify the 200–500 bp regions. Gene-specific primers for *18S rRNA* (X02623) (F 5′-TGATTAACAGGGACAGTCGG-3′; R 5′-ACGGTATCTGATCGTCTTCG-3′) and *nucleoporin* (A0A3B6HV94) (F 5′-GCTGCCTACGAGTCCTTGTC-3′; R 5′-TGATTGAGCAACTTGCCTTG-3′) were synthesized. PCR analysis was performed using the Emerald Amp PCR Master Mix (Takara, Tokyo, Japan) as follows: at 98 °C for 10 s, at 60 °C for 30 s, and at 72 °C for 30 s, for a total of 30 cycles. The amplified products were separated on 3% agarose gel and stained with the Atlas Clear Sight Gold DNA stain (BioAtlas, Tartu, Estonia). The integrated density of bands was calculated using ImageJ software.

### 4.9. Starch-Content Assay

The starch content was analyzed using a Starch Assay Kit (BioAssay Systems, Hayward, CA, USA) according to the protocol of the manufacturer. A portion (10 mg) of samples was ground in phosphate-buffered saline using a mortar and pestle. Free glucose and small oligosaccharides were removed with 1 mL of ethanol, and the homogenate was incubated for 5 min at 60 °C with shaking. After centrifugation at 10,000× *g* for 2 min, the soluble starch in the pellet was extracted with 1 mL of water, incubated for 5 min at 100 °C, and then centrifuged at 10,000× *g* for 2 min. The supernatant was collected as soluble starch. Each sample (10 μL) was added to the working reagent by mixing 90 μL of assay buffer, 2 μL of enzyme, and 1 μL of dye reagent. The mixture was incubated for 30 min at 25 °C, and the absorbance was measured at 570 nm.

### 4.10. Statistical Analysis

Statistical significance between the 2 groups was assessed by the Student’s *t*-test, and a *p*-value of less than 0.05 was considered statistically significant.

## 5. Conclusions

Wheat-root growth suppressed under salt stress was recovered to the control level by MMW irradiation. Proteomic analysis was conducted to reveal the salt-stress tolerance mechanism induced by MMW irradiation in wheat. By identifying specific proteins, proteoforms, and their associated pathways, the mechanisms of MMW irradiation under salt stress were identified in wheat. The following proteins were identified as key proteins: (i) among proteins related to the ROS scavenging system, SOD 1 and GR were accumulated under salt stress and further increased in MMW irradiated wheat; and (ii) among the pathogenesis-related proteins, pathogenesis-related protein 1 and the Bowman–Birk proteinase inhibitor decreased under salt stress but were restored to the control level in MMW-irradiated wheat. These results suggest that the MMW irradiation of wheat seeds enhances plant-growth recovery from salt stress through the modulation of the ROS-scavenging system and stress tolerance. The present results indicate that the MMW irradiation of wheat seeds improves plant-growth recovery from salt stress through the regulation of the reactive oxygen species-scavenging systems and pathogen-related proteins. These genes may contribute to the development of salt-stress tolerant wheat through marker-assisted breeding and genome editing. For this reason, in the future, field verification experiments will likely be necessary.

## Figures and Tables

**Figure 1 ijms-26-00253-f001:**
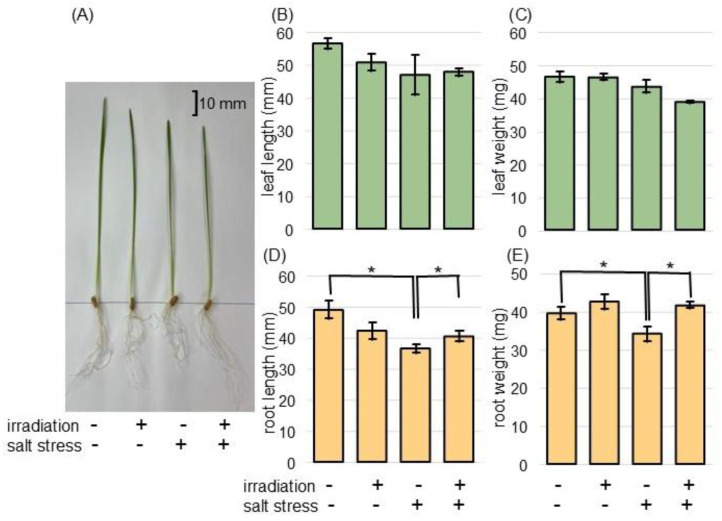
Morphological effects on MMW-irradiated wheat treated with salt stress. Wheat seeds were irradiated with or without MMW and sown. For nontreated groups, wheat seedlings were collected 5 days after sowing (Appendix A). For the salt-stress groups, 3-day-old wheat plants were subjected to salt stress for 2 days and collected (**A**). As seedling collection, leaf (green column) and root (orange column) were collected. As morphological parameters, leaf length (**B**), leaf-fresh weight (**C**), main-root length (**D**), and total-root fresh weight (**E**) were measured. The bar in the picture indicates 10 mm. Data are shown as the mean ± SD from 3 independent biological replicates (Appendix A), with 10 plants per each replicate. Asterisks indicate significant changes between the 2 groups using Student’s *t*-test (* *p* < 0.05).

**Figure 2 ijms-26-00253-f002:**
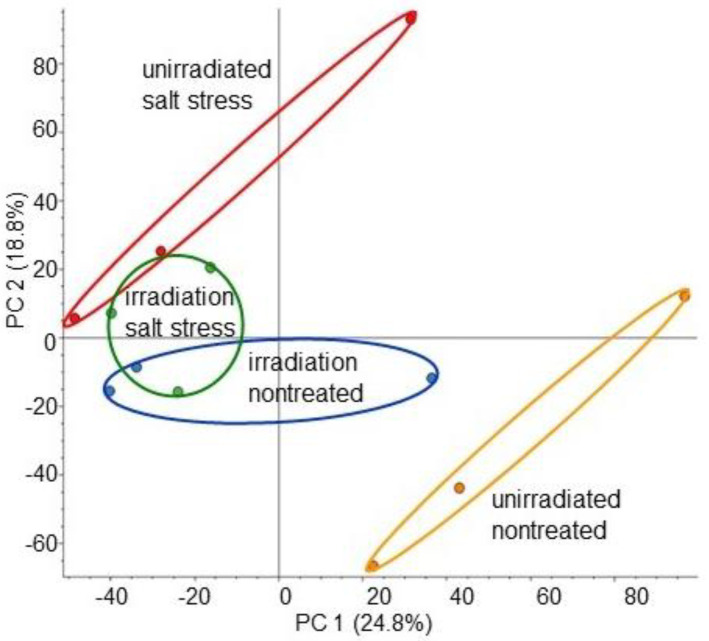
Summary of whole-proteomic data for 12 wheat samples based on principal-component analysis. Wheat seeds irradiated with or without MMW were sown, and the 3-day-old seedlings were treated with or without salt stress for 2 days. Roots from 4 groups were collected, which were unirradiated/nontreated (orange), irradiated/nontreated (blue), unirradiated/salt stress (red), and irradiated/salt stress (green). Proteomics was carried out on three independent biological replicates for each treatment. Principal-component analysis was conducted using Proteome Discoverer.

**Figure 3 ijms-26-00253-f003:**
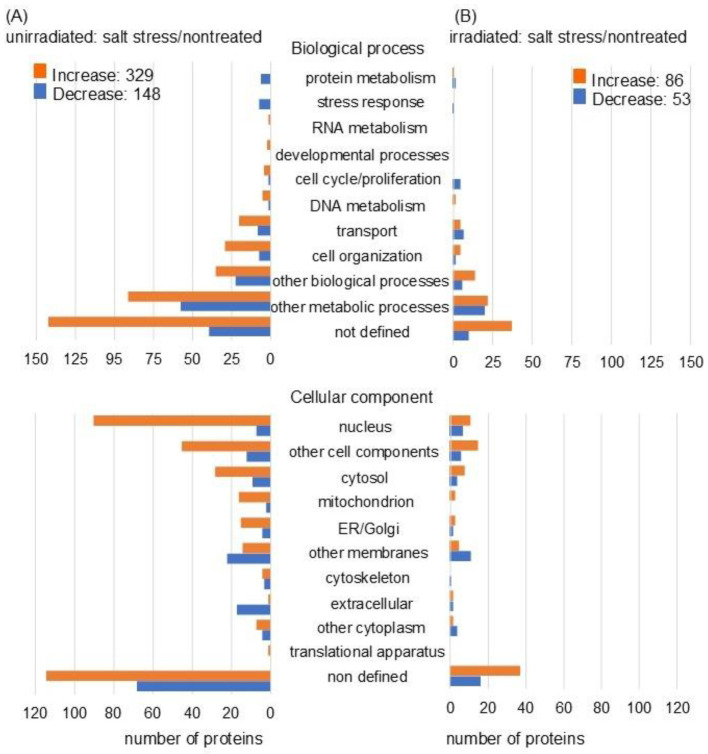
Functional categorization of differentially abundant proteins using proteomic analysis. Wheat seeds were irradiated with (irradiated) or without (unirradiated) MMW, and the seedlings were treated with (salt stress) or without (nontreated) salt stress. After the proteomic analysis of proteins extracted from the roots, functional classification of significantly increased (orange) and decreased (blue) proteins (*p* < 0.05) from unirradiated (**A**) or irradiated (**B**) wheat treated with or without salt stress was performed using gene-ontology analysis (Appendix A).

**Figure 4 ijms-26-00253-f004:**
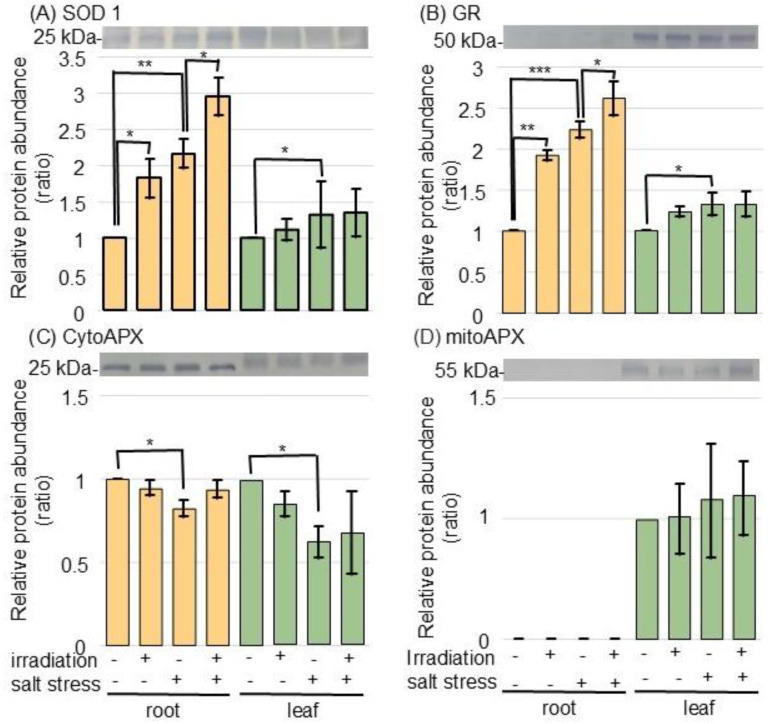
Immunoblot analysis of proteins changed in the ROS-scavenging system of wheat irradiated with MMW under salt stress. Wheat seeds irradiated with or without MMW were sown, and seedlings were treated with or without salt stress. Proteins extracted from the root and leaf of wheat were separated on SDS-polyacrylamide gel by electrophoresis. Coomassie-brilliant blue staining pattern was used as a loading control (Appendix A). Proteins transferred onto membranes were cross-reacted with anti-SOD 1 (Cu/Zn SOD) (**A**), GR (**B**), and APX (**C**,**D**) antibodies. The integrated density of the bands was calculated with ImageJ software. Data are shown as the mean ± SD from 3 independent biological replicates (Appendix A). Statistical analysis is the same as in Figure 1 (*, *p* < 0.05; **, *p* < 0.01; and ***, *p* < 0.001).

**Figure 5 ijms-26-00253-f005:**
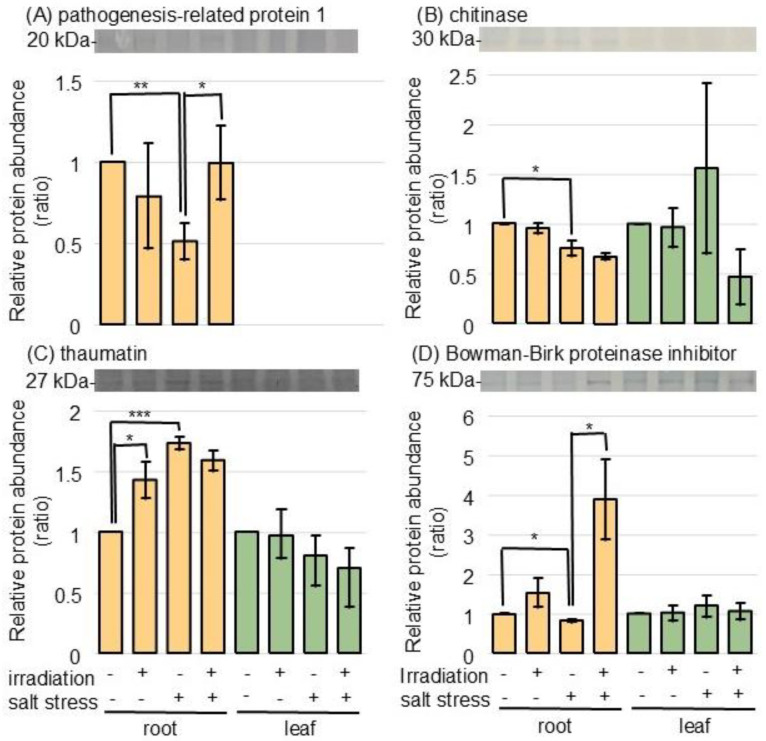
Immunoblot analysis of proteins involved in pathogen resistance in wheat irradiated with MMW under salt stress. The experimental procedures were the same as in Figure 4. The membranes were cross-reacted with anti-pathogenesis-related protein 1 (**A**), chitinase (**B**), thaumatin (**C**), and Bowman–Birk proteinase inhibitor (**D**) antibodies. Data are shown as the mean ± SD from 3 independent biological replicates (Appendix A). The statistical analysis is the same as in Figure 1 (*, *p* < 0.05; **, *p* < 0.01; and ***, *p* < 0.001).

**Figure 6 ijms-26-00253-f006:**
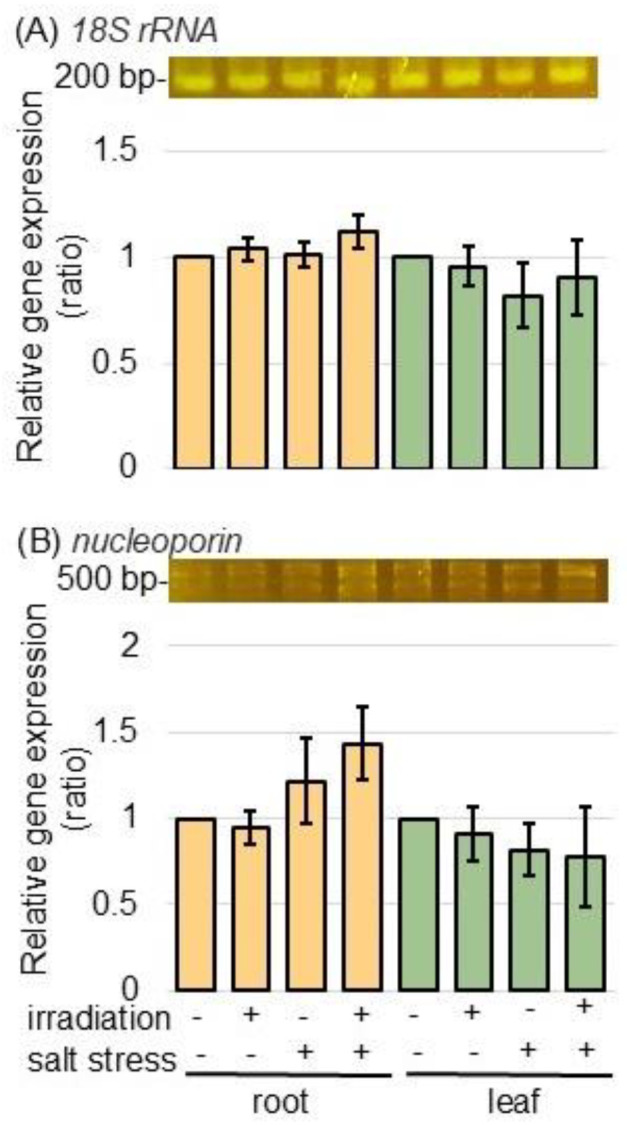
The expression of gene-encoding nucleoporin in wheat irradiated with MMW under salt stress. Wheat seeds were irradiated with or without MMW, and seedlings were treated with or without salt stress. After isolating total RNA from root and leaf samples, *18S rRNA* (**A**)—specific and *nucleoporin* (**B**)—specific oligonucleotides were amplified using PCR. *18S rRNA* was used as an internal control. After agarose-gel electrophoresis, the integrated densities of bands were calculated using ImageJ software. Data are shown as the mean ± SD from 3 independent biological replicates.

**Figure 7 ijms-26-00253-f007:**
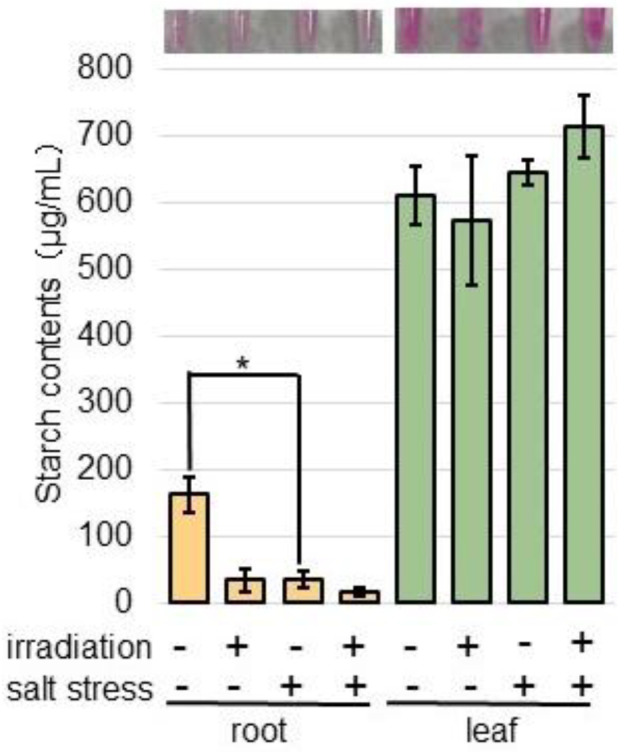
The starch contents in wheat irradiated with MMW under salt stress. Wheat seeds were irradiated with or without MMW, and seedlings were treated with or without salt stress. Starch was extracted from roots and leaves. A portion (10 mg) of samples was ground in 1 mL of phosphate-buffered saline. Free glucose and small oligosaccharides were removed with ethanol. After centrifugation, the soluble starch in the pellet was extracted with 1 mL of water. The contents were analyzed using a Starch Assay Kit. The picture shows the color of the sample after the reaction. Data are shown as the mean ± SD from three independent biological replicates. Statistical analysis is the same as in Figure 1 (*, *p* < 0.05).

## Data Availability

For the MS data, RAW data, peak lists and result files have been deposited in the ProteomeXchange Consortium [64] via the jPOST [65] partner repository under the data-set identifier PXD044428.

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
