# Peer review of "Protein-Based Mechanism of Wheat Growth Under Salt Stress in Seeds Irradiated with Millimeter Waves"

_ijms, 2024, doi:10.3390/ijms26010253_

Round 1

Reviewer 1 Report

Comments and Suggestions for Authors

The paper devoted to study effect of salt and irradiation on adaptation of wheta seedlings.

Authors made a large pieces of work. However, in the current text so much details are missing, authors mix different cell type. And ignore facts that  growth inhibition or sztress response is a results of conflicts between different cell types in one organ. Seed detals below.

Line 15 and more: „even“ – redundant.

Lines 14 – 16: no logic. First you mentioned about recovery, but next you study effects.

Line 17: “largely affected wheat proteins“ ??? It is better to decrobe which proteins here.

Line 40: „Several molecular, biochemical, and physiological mechanisms are involved in the“ ???? Mechanisms can not be invloved itself, Re-formulation need.

Line 48: „were characterized in wheat vacuole“ ?? Gene can not be charaterzed in vacuole!

Lines 66 – 79: please, formulate ckear task andhypothesis here, based in previous descriptions.

Line 100: „cellular mechanism“ ?? In wheat seedlings tehre are so many cell types each have own protein profile and epigenome. And each respond to irradiation and salt differntly. Mixing o fit is not a good idea. The growth inhibition is a results of different response of different cell type (geometrical/epigenomic) conflicts.

Lines 155- 155: The abundance of cytosolic APX decreased with salt stress; however, there was no change with irradiation.“ ?? Iinderstand what do you mean, but please made text more clear.

Why you measure ptotein level? Activity will be moire reasonabale.

Fig 4A – which SOD?

#Line 192: „protein involved in nucleus“ ???  histones? Others? X-axis layout.

Fig 7- unclear. What is ml of  the leaf?

Line 287: are irradiated dry seeds or seeds after soaking? I think soaked seeds will be more effective.

Line 289: „studies [23,24]“ – no details given, but rediricetd tommore eraly work.

Line 290: „20 seeds were sown evenly in each seedling case“?? seeds in seedling case???

Lines 292-297: have you include or exclude merstem (leaf)? Have you incude colepotile?

Line 300: “(500 mg) of samples was ground in 500 μ” ?? The ratio between smaple and buffer shoud be 1:10, not 1:1.

Comments on the Quality of English Language

So many sentences need corrections. Word "even" in almost each sentence.

Author Response

Reviewer 1

The paper devoted to study effect of salt and irradiation on adaptation of wheta seedlings.
Authors made a large pieces of work. However, in the current text so much details are missing, authors mix different cell type. And ignore facts that  growth inhibition or sztress response is a results of conflicts between different cell types in one organ. Seed detals below.
Answer: Thank you very much for your comments. We have made the following corrections based on your comments. We appreciate your comments to improve this paper.

Line 15 and more: „even“ – redundant.
Answer: As suggested, the word “even” has been removed from all parts of the manuscript.

Lines 14 – 16: no logic. First you mentioned about recovery, but next you study effects.
Answer: Thank you very much for your comment. This sentence has been revised as follows: “To reveal the salt-stress tolerance mechanism of MMW irradiation on wheat, proteomic analysis was conducted.” And this problem has been corrected in all parts of the manuscript.

Line 17: “largely affected wheat proteins“ ??? It is better to decrobe which proteins here.
Answer: Thank you very much for your correction. Because protein names were described in the next sentence, this sentence has been removed and the categories of proteins have been added as follows: “Proteins related to cell cycle/proliferation and transport in biological process, as well as proteins related to nucleus, cytoskeleton, and cytoplasm in cellular component, were reversely changed with the number of proteins.”   

Line 40: „Several molecular, biochemical, and physiological mechanisms are involved in the“ ???? Mechanisms can not be invloved itself, Re-formulation need.
Answer: We are sorry about this problem. The sentence has been revised as follows: “Several molecular, biochemical, and physiological changes are involved in the response of crop plants to salt stress”

Line 48: „were characterized in wheat vacuole“ ?? Gene can not be charaterzed in vacuole!
Answer: We are sorry about this problem. This sentence has been corrected as follows: “NHX (Na+/H+ antiporter) group is responsible for vacuolar sequestration of Na+, and 4 proteins encoding NHX genes, including TaNHX1, TaNHX2, TaNHX3, and TaNHX4-B, were characterized in wheat vacuole”

Lines 66 – 79: please, formulate ckear task andhypothesis here, based in previous descriptions.
Answer: As suggested, this paragraph has been revised as follows: “Furthermore, MMW improved plant growth and stress tolerance such as flooding in soybean [20], chickpea [21], and wheat [22]. These results indicated that MMW irradiation could be an effective method to promote plant growth and stress tolerance in crops. To clarify the salt-stress tolerance mechanism of MMW irradiation on wheat, morphological analysis was carried out using irradiated seeds. Because MMW irradiation induced structural changes in proteins, proteomic techniques were performed using roots, whose growth in irradiated wheat was improved under salt stress. This was performed using nano-liquid chromatography (LC) combined with mass spectrometry (MS) to explore the responsible mechanisms for the positive effects of MMW on wheat growth under salt stress. The proteomic results were then confirmed by protein accumulation and other analyses using immunoblot and polymerase-chain reaction (PCR) techniques.”

Line 100: „cellular mechanism“ ?? In wheat seedlings tehre are so many cell types each have own protein profile and epigenome. And each respond to irradiation and salt differntly. Mixing o fit is not a good idea. The growth inhibition is a results of different response of different cell type (geometrical/epigenomic) conflicts.
Answer: Thank you very much for your helpful comments. We now know that the next step is to take more detailed samples. In this stage, the sentence has been revised as follows: “To investigate protein changes induced by MMW irradiation in wheat growth under salt stress, MS-based proteomics was carried out using roots.”

Lines 155- 155: The abundance of cytosolic APX decreased with salt stress; however, there was no change with irradiation.“ ?? Iinderstand what do you mean, but please made text more clear.
Why you measure ptotein level? Activity will be moire reasonabale.
Answer: Thank you for pointing that out. You are right, we should have measured the amount and activity of the protein. We will measure both in the next step. This time, because we did it to verify the results of the proteomics analysis, this point was clearly stated as follows in red: “Because proteomic analysis revealed changes in the amounts of ROS-scavenging and pathogenesis-related proteins, the changes in protein amounts were confirmed by immunoblot analysis (Figures 4 and 5).”

Fig 4A – which SOD?
Answer: We are sorry for not making that point clear. It is SOD 1 (Cu/Zn SOD). Figure 4 and figure legend as well as the contents of manuscript have been revised in red. 

#Line 192: „protein involved in nucleus“ ???  histones? Others? X-axis layout.
Answer: As suggested, this title has been clarified as follows: “The expression of gene encoding nucleoporin in wheat irradiated with MMW under salt stress.” And also thank you very much for your comment. X-axis layout has been revised in Figure 6.

Fig 7- unclear. What is ml of the leaf? 
Answer: We are sorry about this problem. Because the experimental procedure was described in the only experimental methods, the key points have been also listed in figure legends of Figure 7 as follows in red: “A portion (10 mg) of samples was ground in 1 mL of phosphate-buffered saline. Free glucose and small oligosaccharides were removed with ethanol. After centrifugation, the soluble starch in the pellet was extracted with 1 mL of water.”

Line 287: are irradiated dry seeds or seeds after soaking? I think soaked seeds will be more effective.
Answer: Thank you very much for your question. Because MMW is not permeable to water, it cannot be used on soaking seeds. The character of MMW has been added in the introduction section of revised version as follows: “Millimeter waves (MMW) are located at the overlap between infrared and microwaves. Their radio frequencies are between 30 and 300 GHz and wavelengths are between 10 and 1 mm [13]. With long wavelengths, which are not permeable to water and minimal risks to human health, MMW irradiation is a suitable technology for the environment, which has dynamic effects on organisms [14]”.

Line 289: „studies [23,24]“ – no details given, but rediricetd tommore eraly work.
Answer: Thank you very much for your check. The brief description has been included in the section “4.1. Plant Materials and Salt Stress” as follows: “The seeds of wheat (Triticum aestivum L. cultivar Nourin 61) were irradiated with MMW at 20 mW for 20 min. The workflow of MMW irradiation was described in previous studies [23,24] (Table S1). Briefly, a Gunn oscillator was used as a MMW source, with a frequency range of 79 to 115 GHz and output power of 7 to 80 mW. The antenna pattern of the horn antenna had an aperture angle of 17 degrees on each side. By placing a 5 cm diameter petri dish containing wheat seeds at 15 cm from the horn antenna, the MMW radiation area completely covered the dish. The irradiation time and the oscillation power were fixed at 20 mW for 20 min, respectively, which were more effective condition to wheat [22].”

Line 290: „20 seeds were sown evenly in each seedling case“?? seeds in seedling case???
Answer: We are sorry about this problem. This sentence has been revised as follows: “After MMW irradiation, a total of 20 seeds were sown evenly in 400 mL of silica sand in each seedling case.”

Lines 292-297: have you include or exclude merstem (leaf)? Have you incude colepotile?
Answer: We appreciate your helpful comments. In this research, meristems and coleoptiles were excluded. Based on your comments, we plan to use these materials in our future research. 

Line 300: “(500 mg) of samples was ground in 500 μ” ?? The ratio between smaple and buffer shoud be 1:10, not 1:1.to use 
Answer: We appreciate your helpful comments. Because roots contain a large amount of water, sample and buffer volumes were determined as described. However, based on your comments, we will consider the points you have raised in the future.

So many sentences need corrections. Word "even" in almost each sentence.
Answer: We are sorry about this problem. the word “even” has been removed from all parts of the manuscript. Now, this paper has been proofread by an American native English speaker.

Reviewer 2 Report

Comments and Suggestions for Authors

This manuscript explores the effects of millimeter wave (MMW) irradiation on wheat growth under salt stress, with a focus on proteomic analysis to uncover the underlying mechanisms. The research has potential applications in improving crop resilience in saline environments. However, the authors are advised to address the following comments and concerns.

1.      Line 13: The abstract states that "MMW irradiation improved the growth of soybean, chickpea, and wheat under flooding stress." What is the relationship between studying salt stress?

2.      Line 14: Please provide quantitative data or specific percentages to highlight the extent of improvement.

3.      Line 25: At the end of the abstract, please add a summary sentence on how these findings could be applied to improve wheat cultivation under saline conditions.

4.      Line 287: What is the reason for the authors using only one treatment (MMW at 20 mW for 20 minutes)? It would be helpful to explain the rationale behind selecting this specific treatment.

5.      Line 288: The authors notes that the MMW irradiation workflow is detailed in prior studies. It would be helpful to include a brief description here.

6.      Line292: The choice of 100 mM NaCl for salt stress is mentioned, but the rationale is lacking. Justification or standard reference would strengthen the experimental design.

7.      Line 293: Please include more details about the growth conditions and the nutrient solution used.

8.      The introduction is weak, and it fails to clearly articulate the specific research gap that this study aims to address.

9.      The introduction's flow is somewhat disjointed, as it transitions from discussing salt tolerance mechanisms to electromagnetism and millimeter waves (MMW) without establishing a clear connection between them.

10.  Line 83: Provide quantitative data or percentage changes to substantiate claims of significance.

11.  Line 85: Figure 1 appears that the treatment had no effect on leaf length and leaf fresh weight in irradiated wheat under stress. Could you provide an explanation for this observation?

12.  Figure 1B, C, D. Please add letters in state of asterisks.

13.  The discussion briefly mentions the role of ROS scavenging and pathogen-related proteins in salt-stress tolerance but lacks depth. Providing a detailed explanation or hypothesis would add clarity.

14.  The conclusions should be re-written to avoid reiterating details from the results and discussion. A concise summary emphasizing broader implications, such as improving crop resilience in saline soils, and addressing limitations like the need for field trials would enhance their impact.

Comments on the Quality of English Language

The English could be improved to more clearly express the research.

Author Response

Reviewer 2   This manuscript explores the effects of millimeter wave (MMW) irradiation on wheat growth under salt stress, with a focus on proteomic analysis to uncover the underlying mechanisms. The research has potential applications in improving crop resilience in saline environments. However, the authors are advised to address the following comments and concerns. Answer: Thank you very much for your comments. We have made the following corrections based on your comments. We appreciate your comments to improve this paper.   1.      Line 13: The abstract states that "MMW irradiation improved the growth of soybean, chickpea, and wheat under flooding stress." What is the relationship between studying salt stress? Answer: Thank you very much for your comments. This sentence has been revised as follows: “Furthermore, MMW improved the plant growth and stress tolerance such as flooding in soybean [20], chickpea [21], and wheat [22]. These results indicated that MMW irradiation could be an effective method to promote plant growth and stress tolerance in crops.”   2.      Line 14: Please provide quantitative data or specific percentages to highlight the extent of improvement. Answer: We are sorry about this problem. The sentences in the abstract section and result section as well as discussion section have been revised in red as follows: “In the present study, wheat-root growth suppressed to 77.6% of the control by salt stress was recovered to the control level by MMW irradiation.”   3.      Line 25: At the end of the abstract, please add a summary sentence on how these findings could be applied to improve wheat cultivation under saline conditions. Answer: Thank you very much for your useful comments. As suggested, a future perspective was added at the end of the abstract in red as follows: “The present results indicate that MMW irradiation of wheat seeds improves plant-growth recovery from salt stress through regulating reactive-oxygen species scavenging system and pathogen related proteins. These genes may contribute to the development of salt-stress tolerant wheat through marker-assisted breeding and genome editing.”   4.      Line 287: What is the reason for the authors using only one treatment (MMW at 20 mW for 20 minutes)? It would be helpful to explain the rationale behind selecting this specific treatment. Answer: We are sorry for the lack of explanation. The explanation of the rationale behind selecting this specific treatment has been added as follows:  “The irradiation time and the oscillation power were fixed at 20 mW for 20 min, respectively, which were more effective condition to wheat [22].”   5.      Line 288: The authors notes that the MMW irradiation workflow is detailed in prior studies. It would be helpful to include a brief description here. Answer: As suggested, the brief description has been included in the section “4.1. Plant Materials and Salt Stress” as follows: “The seeds of wheat (Triticum aestivum L. cultivar Nourin 61) were irradiated with MMW at 20 mW for 20 min. The workflow of MMW irradiation was described in previous studies [23,24] (Table S1). Briefly, a Gunn oscillator was used as a MMW source, with a frequency range of 79 to 115 GHz and output power of 7 to 80 mW. The antenna pattern of the horn antenna had an aperture angle of 17 degrees on each side. By placing a 5 cm diameter petri dish containing wheat seeds at 15 cm from the horn antenna, the MMW radiation area completely covered the dish. The irradiation time and the oscillation power were fixed at 20 mW for 20 min, respectively, which were more effective condition to wheat [22].”   6.      Line292: The choice of 100 mM NaCl for salt stress is mentioned, but the rationale is lacking. Justification or standard reference would strengthen the experimental design. Answer: Thank you very much for your comments. The reason has been added as follows: “Furthermore, based on previous study, a growth-inhibiting concentration of NaCl for wheat, which is 100 mM, was used [Komatsu et al., 2024]. In this study, to induce salt stress, 100 mM NaCl were supplied for 2 days.”   7.      Line 293: Please include more details about the growth conditions and the nutrient solution used. Answer: Since the wheat used was five days after sowing, no nutrient solution was used in this experiment. The growth condition has been revised in the section “4.1. Plant Materials and Salt Treatment” as follows: “After MMW irradiation, a total of 20 seeds were sown evenly in 400 mL of silica sand in each seedling case. After it, they were cultivated for 3 days at 25°C under white-fluorescent light (160 µmol m-2 s-1, 12 h light period/day).”   8.      The introduction is weak, and it fails to clearly articulate the specific research gap that this study aims to address. 9.      The introduction's flow is somewhat disjointed, as it transitions from discussing salt tolerance mechanisms to electromagnetism and millimeter waves (MMW) without establishing a clear connection between them. Answer: We thought numbers 8 and 9 were similar criticisms, so we fixed them together.  We appreciate your comment. The third and fourth paragraphs of the Introduction have been significantly rewritten in red. We hope the current paragraph is clearer.   10.  Line 83: Provide quantitative data or percentage changes to substantiate claims of significance. Answer: Thank you very much for your request. Quantitative data have been added as new supplemental Table 2 and percentage change has been added in the result section in red.   11.  Line 85: Figure 1 appears that the treatment had no effect on leaf length and leaf fresh weight in irradiated wheat under stress. Could you provide an explanation for this observation? Answer: Thank you very much for your suggestion. In this experiment, no nutrients were added, so even though the roots grew better, the nutrients were not absorbed, so there may have been no effect on the above-ground parts. This explanation has been added to the discussion section. Additionally, the sentence of result section has been corrected as follows: “However, based on statistical analysis by t-test, no significant changes were observed in fresh weight and length of leaf under salt stress in both unirradiated and irradiated wheat.”   12.  Figure 1B, C, D. Please add letters in state of asterisks. Answer: The explanation of figure legends of Figure 1 has been revised as follows: “As seedling collection, leaf (green column) and root (orange column) were collected. As morphological parameters, leaf length (B), leaf-fresh weight (C), main-root length (D), and total-root fresh weight (E) were measured.”  Additionally, an explanation of Asterisks has been added to the figure legend as follows: “Asterisks indicate significant changes between the 2 groups by Student’s t-test (*p < 0.05).”   13.  The discussion briefly mentions the role of ROS scavenging and pathogen-related proteins in salt-stress tolerance but lacks depth. Providing a detailed explanation or hypothesis would add clarity. Answer: Thank you very much for your useful comments and suggestions. For each paragraph, a detailed explanation or hypothesis has been added in red.   14.  The conclusions should be re-written to avoid reiterating details from the results and discussion. A concise summary emphasizing broader implications, such as improving crop resilience in saline soils, and addressing limitations like the need for field trials would enhance their impact. Answer: Thank you very much for your comments. As suggested, the conclusion section has been revised with the need for field trials. The corrected parts in the conclusion section have been marked in red.   The English could be improved to more clearly express the research. Answer: We are sorry about this problem. Now, this paper has been proofread by an American native English speaker.

Round 2

Reviewer 1 Report

Comments and Suggestions for Authors

Thank you! The text is OK now. However, as a see, many sentences require slightly polishing.

Please, ask external person to do this.

My best regards!

Happy New Year!

Reviewer 2 Report

Comments and Suggestions for Authors

The authors have successfully addressed most of the suggestions, significantly improving the quality and clarity of the manuscript.